# Excess Mortality by Multimorbidity, Socioeconomic, and Healthcare Factors, amongst Patients Diagnosed with Diffuse Large B-Cell or Follicular Lymphoma in England

**DOI:** 10.3390/cancers13225805

**Published:** 2021-11-19

**Authors:** Matthew James Smith, Aurélien Belot, Matteo Quartagno, Miguel Angel Luque Fernandez, Audrey Bonaventure, Susan Gachau, Sara Benitez Majano, Bernard Rachet, Edmund Njeru Njagi

**Affiliations:** 1Inequalities in Cancer Outcomes Network, Department of Non-Communicable Disease Epidemiology, London School of Hygiene and Tropical Medicine, London WC1E 7HT, UK; aurelien.belot@lshtm.ac.uk (A.B.); miguel-angel.luque@lshtm.ac.uk (M.A.L.F.); Sara.benitezmajano@lshtm.ac.uk (S.B.M.); bernard.rachet@lshtm.ac.uk (B.R.); edmund.njeru.njagi@lshtm.ac.uk (E.N.N.); 2MRC Clinical Trials Unit, Institute of Clinical Trials and Methodology, University College London, London WC1V 6LJ, UK; m.quartagno@ucl.ac.uk; 3Noncommunicable Disease and Cancer Epidemiology Group, Instituto de Investigación Biosanitaria de Granada, Ibs.GRANADA, Andalusian School of Public Health, 18012 Granada, Spain; 4Centro de Investigación Biomédica en Red de Epidemiología y Salud Pública (CIBER of Epidemiology and Public Health, CIBERESP), 28029 Madrid, Spain; 5Epidemiology of Childhood and Adolescent Cancers Team, Research Centre in Epidemiology and Biostatistics (CRESS), Inserm UMR 1153, Université de Paris, 94801 Villejuif, France; audrey.bonaventure@inserm.fr; 6School of Mathematics, University of Nairobi, Nairobi 30197-00100, Kenya; sgachau06@gmail.com

**Keywords:** cancer epidemiology, diffuse large B-cell lymphoma, follicular lymphoma, survival analysis, comorbidity, multimorbidity, socioeconomic status

## Abstract

**Simple Summary:**

Diffuse large B-cell (DLBCL) and follicular lymphoma (FL) account for most non-Hodgkin lymphoma diagnoses: around 35% and 20% in England, respectively. Despite the vast contrast in survival between the subtypes, similar socioeconomic inequalities in survival have persisted over the past two decades, possibly due to the presence of comorbidities. The aim of our study was to assess the association between socioeconomic status and survival from DLBCL or FL accounting for patient and health system characteristics. We found that, for both DLBCL and FL, the most deprived patients had a higher excess mortality hazard compared to the least deprived, regardless of the comorbidity status. Our results show the need for the current framework of the National Health Service to improve the survival of DLBCL and FL patients in the most deprived areas of England, and further consideration is needed for patient-tailored management plans amongst patients with comorbidities or multimorbidities.

**Abstract:**

(1) Background: Socioeconomic inequalities of survival in patients with lymphoma persist, which may be explained by patients’ comorbidities. We aimed to assess the association between comorbidities and the survival of patients diagnosed with diffuse large B-cell (DLBCL) or follicular lymphoma (FL) in England accounting for other socio-demographic characteristics. (2) Methods: Population-based cancer registry data were linked to Hospital Episode Statistics. We used a flexible multilevel excess hazard model to estimate excess mortality and net survival by patient’s comorbidity status, adjusted for sociodemographic, economic, and healthcare factors, and accounting for the patient’s area of residence. We used the latent normal joint modelling multiple imputation approach for missing data. (3) Results: Overall, 15,516 and 29,898 patients were diagnosed with FL and DLBCL in England between 2005 and 2013, respectively. Amongst DLBCL and FL patients, respectively, those in the most deprived areas showed 1.22 (95% confidence interval (CI): 1.18–1.27) and 1.45 (95% CI: 1.30–1.62) times higher excess mortality hazard compared to those in the least deprived areas, adjusted for comorbidity status, age at diagnosis, sex, ethnicity, and route to diagnosis. (4) Conclusions: Deprivation is consistently associated with poorer survival among patients diagnosed with DLBCL or FL, after adjusting for co/multimorbidities. Comorbidities and multimorbidities need to be considered when planning public health interventions targeting haematological malignancies in England.

## 1. Introduction

Non-Hodgkin lymphoma (NHL) is a heterogeneous group of malignancies, and is currently the 6th most commonly diagnosed cancer in England; in 2014, approximately 32 males and 23 females per 100,000 person years were diagnosed [1]. The heterogeneity in morphology leads to variation in survival probability; for instance, 5-year survival of follicular lymphoma (FL) (86.3%) is higher than diffuse large B-cell lymphoma (DLBCL) (54.8%) [2].

The healthcare system in England aims to offer equitable access to care for all patients. However, variability in health outcomes amongst patients with similar cancers and sociodemographic characteristics still occur; [2,3,4] convincing reasons for this variability remain a topic of interest. In 2001, the National Health Service (NHS) Cancer Plan [5] recognised, and aimed to reduce, the disparities in survival. Since implementation, there is no evidence that the Plan has had an impact on the inequalities [6,7]. The deprivation gap in survival is still apparent, despite the Plan and successive policies [5,8,9,10], illustrating the incomplete understanding of the mechanisms underlying these inequalities and raising the concern that these policies have missed the relevant targets.

Patients’ comorbidity status may impact timely diagnosis, possibly leading to treatment with more adverse effects [11]; comorbidities are, on average, more prevalent and severe amongst more deprived patients [12]. However, recent evidence indicates that comorbidity explains little of the differential cancer survival between socioeconomic groups [13,14,15]. Variations in healthcare access, such as location of residence, could partly explain the inequalities [16,17,18,19,20].

Since population-based cancer registries rarely hold reliable information on the cause of death, cancer-specific mortality estimates can be estimated with relative survival methods. These methods compare the mortality hazard (i.e., excess mortality hazard) observed in a population of cancer patients to the mortality hazard observed in the general population with identical demographic characteristics. In this context, the survival estimate derived from the excess mortality hazard is termed net survival (or cancer survival), which is interpreted as the survival where death is due directly, or indirectly, to the cancer studied, and death from other causes has been removed [21].

Overall, the association between comorbidity and cancer survival in patients with DLBCL and FL, accounting for other socio-demographic characteristics and the area of residence, remains unclear. We aim to describe the association between comorbidities and cancer survival amongst DLBCL or FL patients, while accounting for sociodemographic and economic factors, hypothesizing that the presence of comorbidities is associated with poorer survival.

## 2. Methods

### 2.1. Study Design, Participants, and Data Sources

We developed a population-based multilevel cohort study of adult patients diagnosed with DLBCL or FL between 1 January 2005 and 31 December 2013 in England. Patients were followed up until death or the end of the study on the 31 December 2015, whichever occurred first.

DLBCL and FL were defined according to the 10th revision of the International Statistical Classification of Diseases and Related Problems (ICD-10 codes C82.0–C85.9) [22]. Morphology (cell type) and topography (tumour site) were defined using renewed updates of the ICD for Oncology (ICD-O); ICD-O-3 [23] was used for diagnoses up to 2010, and ICD-O-3.1 [24] for diagnoses after 2011. Information on patients with DLBCL or FL was collected from the linkage of English cancer registry data, the Cancer Analysis System [25] (CAS), and Hospital Episode Statistics [26] (HES) data sets within the National Cancer Registry and Analysis Service (NCRAS). These datasets contained detailed information on patient and tumour characteristics (see details below).

### 2.2. Outcome, Exposure, and Patients’ Sociodemographic Characteristics

The outcome of the study was the time to death, or censoring, among DLBCL and FL patients 5 years after cancer diagnosis. Net survival was deduced after estimating the excess mortality hazard. Hence, we used England life tables stratified by deprivation, sex, age, and calendar year (2005–2013) to account for the overall mortality rate from the background population [27]. As follow up of patients ended in 2015 and life tables were available until 2013, we assumed that the expected mortality rates plateaued for 2014 and 2015.

Comorbidity status was the main exposure. We defined comorbidity as the existence of other chronic medical disorders, in addition to cancer, the primary disease of interest, which are causally unrelated to the primary disease [28,29]. Records from HES were used to identify patients’ comorbidity status based on a computational algorithm published elsewhere [30]. The algorithm searches for the presence of comorbidities retrospectively and defines a time window of 6 to 24 months prior to cancer diagnosis where comorbidities are recorded to avoid bias due to the presence of comorbidities related to cancer (i.e., cardiological comorbidities due to DLBCL or FL cancer treatment). Patient comorbidity status was adapted from the original Charlson comorbidity index [31] (CCI). We used the Royal College of Surgeons (RCS) modified Charlson Score (Table A1) [32]. The score removes patients with a previous malignancy to avoid bias, does not assign different weights to comorbidities, and categorises comorbidities as: no comorbidities, one comorbidity, and two or more comorbidities (multimorbidity).

Socio-demographic and economic characteristics were collected from the HES dataset. Age was specified at time of diagnosis. Sex is recorded as male or female. Ethnicity was recorded as white or other. Area-level deprivation, classified into one of five quintiles, was determined by the Index of Multiple Deprivation [33] (IMD), which was based on the Lower Super Output Area [34] (LSOA) residence of the patient at the time of cancer diagnosis. LSOA is a geographical location with a median of 1500 inhabitants. We also include the information regarding patients’ diagnosis path (route to diagnosis), a UK-specific programme, classified as: accident and emergency room diagnosis, general practitioner referral (routine and urgent referrals where the patient was not referred under two-week-wait), two-week-wait (urgent GP referral with a suspicion of cancer), and secondary care diagnosis (other outpatient and inpatient elective routes) [35].

### 2.3. Statistical Analysis

We tabulated the sociodemographic characteristics by DLBCL and FL. To estimate the excess mortality hazard, we used a multilevel excess hazard regression model (EHM) with a cubic B-spline with two knots placed at 1 and 3 years after diagnosis for the baseline hazard $\lambda_{0}\left(t\right)$. We accounted for the hierarchical structure of the data via the inclusion of a random effect [36]. The statistical contribution of the random effect to the overall goodness of fit of the model was tested using a likelihood ratio test statistic with a Chi-square mixture distribution [37]. From the estimated excess hazard, we could deduce the net survival via the classical relationship between hazard and survival [38]. Net survival is the survival associated with the cancer under study, after eliminating the other causes of death.

In the EHM we included the following variables: age, sex, comorbidities (categorical, 3 categories), deprivation (categorical, 5 categories), lymphoma subtype, ethnicity, and route of cancer diagnosis. We included the non-linear effect of age using a regression spline (defined using a truncated power basis) with one knot located at 70 years of age. Furthermore, we assumed a time-dependent effect of age at diagnosis, represented by the interaction between B-spline function of time and age. The parameter estimates for the variables were interpreted conditionally on the random effect, i.e., they have a cluster-specific interpretation, where a cluster refers to a given LSOA. From the model we derived the excess mortality hazard ratios (EMHR) and their respective 95% confidence intervals (CI) for all the categorical variables, and the variance of the random effect for the LSOA. Empirical Bayes estimates of the random effect were used to explore the between-LSOA variability in the excess mortality hazard from DLBCL or FL. The random effect was tested for using a likelihood ratio test, with the reference distribution being a mixture of chi-squared distributions with 0 and 1 degrees of freedom, to account for the well-known boundary problem for random effect variances [39,40].

### 2.4. Missing Data Analysis

We explored the missing data mechanism for the two variables with missing data (ethnicity (FL 24.9%, DLBCL: 22.7%) and route (FL: 7.8%, DLBCL 5.0%)). Due to clustered data and partially observed categorical variables, we used the latent normal joint modelling multiple imputation approach, under a missing at random assumption (MAR) [41]. The imputation model included all fully and partially observed variables, vital status indicator, and the Nelson–Aalen estimate of the cumulative overall hazard, and accounted for clustering of patients within lower super output areas. We generated 10 imputed datasets. The multilevel EHM was fitted to each of these datasets, and results combined using Rubin’s rules [42,43]. Overall tests for the effects of age after multiple imputation were carried out using the F-based procedure for the test of multiple parameters after multiple imputation [41].

We used R software (version 4.1.2, R Development Core Team, 2020, R: *A language and environment for statistical computing*, R Foundation for Statistical Computing, Vienna, Austria) for all data analyses; the mexhaz [36] package was used for excess hazard modelling and the jomo [44] package for multiple imputation.

## 3. Results

Overall, 15,516 (34.2%) patients were diagnosed with FL and 29,898 (65.8%) diagnosed with DLBCL in England between 2005 and 2013 (Table 1). The prevalence of at least one comorbidity was higher amongst DLBCL (10.7%) compared to FL (7.5%). The average age was lower amongst FL compared to DLBCL, 63.9 compared to 67.4 years, respectively. The prevalence of DLBCL was higher amongst deprived areas (16.0%) than FL (14.4%). ‘White’ was the most prevalent ethnicity for both FL (94.9%) and DLBCL (94.1%). GP referral was the most common route to diagnosis amongst FL (44.0%), whereas amongst DLBCL, A&E was the most common (33.8%).

In the multivariable analysis (Table 2), amongst DLBCL, and after multiple imputation, patients with comorbidity and multimorbidity showed 23% and 40% increased excess mortality compared to patients without comorbidity (i.e., EMHR: 1.23; 95% CI: 1.14–1.32, and EMHR: 1.40; CI: 1.01–1.94, respectively). Patients living in the most deprived areas had 1.22 (95% CI: 1.18–1.27) times higher excess mortality than those living in the least deprived areas. Patients diagnosed through A&E had nearly three times a higher excess mortality compared to GP referral (i.e., EMHR: 2.75; 95% CI: 2.54–2.98). Females had a significantly lower excess mortality compared to males (i.e., EMHR 0.93; 95% CI: 0.90–0.96). There was, however, no evidence of a difference in excess mortality by ethnicity (Table 2). Using a likelihood ratio test (a mixture of chi-square distributions) there was strong evidence (*p* < 0.001) that including the random effect improved the fit of the model.

In the multivariable analysis (Table 3), amongst FL, patients with comorbidity and multimorbidity showed 1.52 and 2.19 times the excess mortality compared to patients without comorbidity (i.e., EMHR: 1.52; 95% CI: 1.25–1.84, and EMHR: 2.19; CI: 1.45–3.31, respectively). Patients living in the most deprived areas had 1.45 (95% CI: 1.30–1.62) times higher excess mortality than those living in the least deprived areas. Patients diagnosed through A&E had nearly three times a higher excess mortality compared to GP referral (i.e., EMHR: 3.32; 95% CI: 2.49–4.43). Females had a significantly lower excess mortality compared to males (i.e., EMHR 0.89; 95% CI: 0.81–0.97). There was, however, no evidence of a difference in excess mortality by ethnicity (Table 3). Using a likelihood ratio test (a mixture of chi-square distributions), there was strong evidence (*p* < 0.001) that including the random effect improved the fit of the model.

Figure 1 and Figure 2 show the EMHR for patients with DLBCL and FL, respectively, according to age at diagnosis at different time since diagnosis (Figure 1A and Figure 2A), and according to time since diagnosis for different age at diagnosis (Figure 1B and Figure 2B). The excess mortality hazard for DLBCL and FL patients for different values of age at diagnosis is shown in the Appendix A (Figure A1 and Figure A2, respectively). These plots were obtained from the three-dimensional plots of EMHR, as shown in the Appendix A (Figure A3 and Figure A4, respectively). For DLBCL (Figure 1), the EMHR was higher for older patients whatever the follow-up time (Figure 1A). For those of older or younger ages, in comparison to 70-year-olds, the EMHR was markedly different immediately after, or at 5 years since, diagnosis, but was most similar around 18 months after diagnosis (Figure 1B).

For FL (Figure 2), the non-linear effect of age was almost similar whatever the time since diagnosis; being older was associated with a higher excess mortality hazard (Figure 2A). For those of older or younger ages, in comparison to 70-year-olds, the EMHR was markedly different immediately after or at 5 years since diagnosis but was most similar around 18 months after diagnosis (Figure 2B).

Figure 3 and Figure 4 show the net survival probability as predicted from the regression model amongst patients with DLBCL and FL, respectively. Amongst DLBCL patients (Figure 3), those living in more deprived areas experienced approximately 7% lower 5-year survival compared to patients in the least deprived areas (e.g., 5-year net survival, amongst those without comorbidities, was 56% for the least deprived compared to 49% for the most deprived). Amongst FL patients (Figure 4), those living in more deprived areas experienced approximately 4% lower 5-year survival compared to those living in the least deprived areas (e.g., 5-year net survival, amongst those without comorbidities, was 86% for the least deprived compared to 82% for the most deprived). For DLBCL only (Figure 3), the deprivation gap in survival was apparent from approximately 6 months after diagnosis, regardless of the comorbidity status.

In the Appendix A, we graphically illustrate the empirical Bayes (EB) estimates of the LSOA random effect for the excess mortality hazard from DLBCL and FL (Figure A5 and Figure A6, respectively). A positive EB estimate indicated a higher excess mortality hazard for a patient from that LSOA in comparison with a patient who has similar observed characteristics but from an LSOA with either a less positive, or negative EB estimate. The EB estimates were grouped by deprivation level, to which the LSOA contributed. For both DLBCL and FL (Figure A5 and Figure A6, respectively), the results show there were no outliers and approximately equal distribution of the EB estimates for each deprivation level.

## 4. Discussion

We found strong evidence of a higher excess mortality amongst DLBCL and FL patients diagnosed with comorbidities compared to patients without comorbidities after adjusting for age, deprivation level, ethnicity, and route to diagnosis and accounting for the patient’s area of residence; we also found a noticeable deprivation gap in cancer survival.

Differences in access to treatments, or risk of adverse effects, may explain some of the disparities in survival among DLBCL patients. Immunotherapy (rituximab) for the treatment of aggressive lymphomas (e.g., DLBCL) is known to be effective for those of an advanced age [45,46,47]. Rituximab is often used in combination with doxorubicin, an increase in dosage of which is associated with an increased incidence of adverse effects (cardiotoxicity), such as congestive heart failure [48]. Guidelines based on National Institute for Health and Care Excellence (NICE) recommend that patients at risk of cardiotoxicity, or low tolerance of intensive therapy, consider a less-intensive treatment regimen [49,50,51]. This less-intensive treatment allocation may partly explain the comorbidity inequalities in survival from DLBCL. For patients with FL, the standard management is ‘watch-and-wait’; thus, in the absence of a treatment, the comorbidity inequalities in survival may be largely explained by the presence of a comorbidity itself rather than being explained by the effect of comorbidity on treatment. Novel treatment strategies are being explored to ascertain the survival and quality of life benefits in comparison to current standards of care [52]; however, access to these treatments will depend on (i) the specialist centre available to the patient, and (ii) the patients’ influence on the decision of a treatment allocation. While the results of clinical trials provide insights into the efficacy of a treatment, they may lack external validity when the treatment is administered within the healthcare system. Pragmatic trials could be developed to understand the real-world benefit of NHL treatments, and how they are delivered, to patients of older ages or with underlying health conditions who would often not be eligible for clinical trials [53]. In addition, real-world data are becoming increasingly more available and would be of great utility to evaluate the effectiveness of treatment on the whole population [54,55].

For FL patients, we showed that the excess mortality hazard among older patients compared to younger patients is highest after just 4 years since diagnosis (Figure 2B). Since we accounted for background population mortality, and adjusted for comorbidity, the higher excess hazard could be because of histological transformation from lower to higher grades of FL. Studies suggest the risk of histological transformation increases by 3% per year since cancer onset [56]. Thus, the increased excess hazard amongst older patients may be because histological transformation, which complicates the treatment and management of FL. The effects of time-varying variables (e.g., progression/relapse, treatment response, transplant, etc.) on the excess mortality hazard would be strong predictive factors, but this information was not available from the data. This represents an interesting topic for further investigation, where we could either use an extension of the flexible parametric models for time-dependent variables, or a landmark approach [57].

The importance of understanding the association of comorbid conditions with cancer patients’ outcomes has been well documented [58]. To our knowledge, this is the first study of England cancer registry data investigating survival by comorbidity status among DLBCL and FL patients. Our results are consistent with previous findings from a Danish study, which showed that the hazard of death increased with severity of comorbidity status [59]; however, the study did not account for missing data and the association with comorbidities was potentially overestimated. The EHR associated with comorbidity decreased after accounting for missing data. The deprivation gap in survival persists even after accounting for prognostic factors such as comorbidity [59,60,61]. Smith et al. [3] reported no deprivation gap in survival; however, their study may have lacked power, and their study used the relative survival ratio, which can be biased over longer-term follow up [62].

Consistent with previous studies [4], survival after GP referral (non-emergency) diagnosis is significantly better compared with A&E. However, our study also finds that patients diagnosed through TWW, who would be expected to have worse symptoms and survival, showed no evidence of a difference in survival compared with GP referral. There are two possible reasons for the absence of a difference in the associations. Firstly, GPs could advocate for a prompt referral even though the patient is not on the TWW pathway, resulting in patients with similar access to healthcare facilities. Secondly, on the other hand, patients referred through the TWW pathway have more severe symptoms and are expected to have a higher excess hazard. Our results show no difference in the excess mortality, indicating that the TWW pathway prevents patients with more severe symptoms from having a higher excess hazard. This suggests that the performance of TWW pathway is at least as beneficial to a patient’s survival as GP referral. Other studies have suggested ways to improve outcomes for patients diagnosed with comorbidities, which include: novel treatment strategies [63], the inclusion of elderly patients in clinical trials [64,65], and the investigation of dose allocation amongst those with higher comorbidity scores [66]. However, further factors associated with the interactions between comorbidities and health care systems leading to poorer survival among DLBCL and FL cancer patients need to be studied.

The strengths of this study are that, firstly, we used a large population-based sample size obtained from cancer registry databases linked to HES, which encompasses all patients in England with a diagnosis of DLBCL and FL between 2005 and 2013. HES data encapsulate a national coverage of comorbidities diagnosed during hospital admission and may have missed comorbidities diagnosed during primary care (e.g., diabetes diagnosed during a GP consultation). However, the addition of information provided from comorbidity records captured during primary care may not improve the prediction of cancer patient survival beyond what is captured in HES data [67]. For example, information on comorbidities, such as diabetes, diagnosed outside of hospital admission are likely to have a minimal impact on the prediction of survival beyond information captured in HES. Secondly, we used the Royal College of Surgeons’ adaptation [32] of the Charlson comorbidity score, which provides a more valid measure of the patient’s comorbidity status, because it was developed within the England population healthcare data setting. Thirdly, we used a latent normal joint modelling multiple imputation to treat missing data in ethnicity and diagnostic route. This approach allows the imputation of a mix of variable types, while accounting for multilevel structures arising from the clustering of patients within LSOAs [41,68,69]. We assumed that missing data on partially observed variables were missing at random, given the observed variables; further analysis could explore the violation of this assumption and impute under a not-missing-at-random assumption.

This study has its limitations. Firstly, individual-level socioeconomic measures are recommended in addition to area-level measures [70]. Information on individual-level socioeconomic status was unavailable, but using area-level measures captures the multidimensional composition of a patient’s deprivation level, in addition to the contextual level [33,71]. Furthermore, using area-level measures, there is greater consistency in the measurement of deprivation between time periods, because deprivation scores have a high concordance amongst updates [33]. Secondly, due to data availability, we did not include tumour stage, which may have partly explained the socioeconomic inequalities in survival. However, even though reliable estimates can be obtained after the multiple imputation of partially observed variables with high proportions of missing data [72], the inclusion of tumour stage may not have provided further information for the prediction of survival beyond that of diagnostic route, because late cancer stage is strongly associated with delayed diagnostic route [73].

Survival at 1 and 5 years since diagnosis of DLBCL and FL in England trails that of other European countries [74]; however, restricting estimates to those surviving at least 1 year after diagnosis (conditional survival) shows a comparable 5-year survival [75]. This indicates that long-term survival differences are largely explained by the increased short-term mortality. Understanding long-term survival from FL is more complex due to the histological transformation of indolent lymphomas, which would require an adaptation of the treatment, support, and management from healthcare facilities. This adaptation could be compounded by the patient’s susceptibility to cardiotoxic treatments. Further studies could focus on the mechanisms and inequalities of short-term mortality, the long-term survival of patients with transformed lymphomas, and the survival of patients at risk of cardiotoxicity.

## 5. Conclusions

After accounting for sociodemographic factors, healthcare factors, socioeconomic deprivation, and the patient’s area of residence, comorbidities were consistently associated with poorer survival and an increased excess mortality amongst patients with DLBCL or FL in England. Furthermore, survival inequalities between socioeconomic levels in patients with DLBCL or FL persisted after accounting for the presence of comorbidities and multimorbidities. These results show the need for the current framework of the National Health Service to improve the survival of DLBCL and FL patients in the most deprived areas of England, and further consideration is needed for patient-tailored management plans amongst patients with comorbidities or multimorbidities.

## Figures and Tables

**Figure 1 cancers-13-05805-f001:**
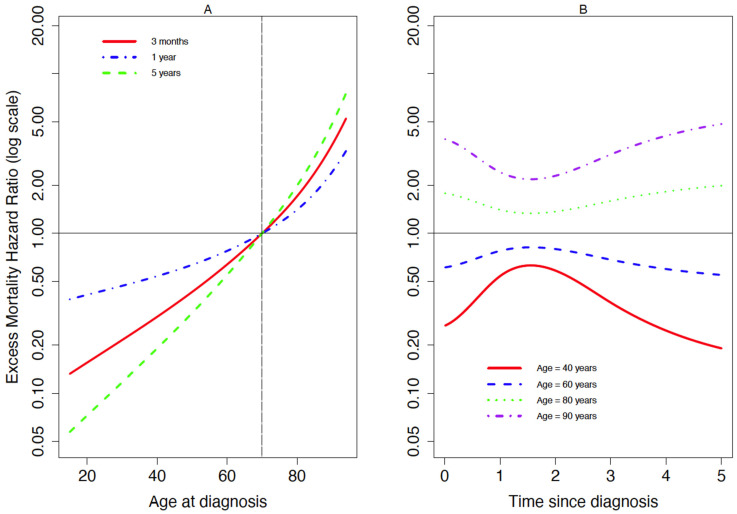
Excess mortality hazard ratios according to (**A**) age at diagnosis at different time since diagnosis (3 months, 1 and 5 years), and (**B**) time since diagnosis for different age groups, amongst patients diagnosed with diffuse large B-cell lymphoma (*n* = 29,898) in England during 2005–2013.

**Figure 2 cancers-13-05805-f002:**
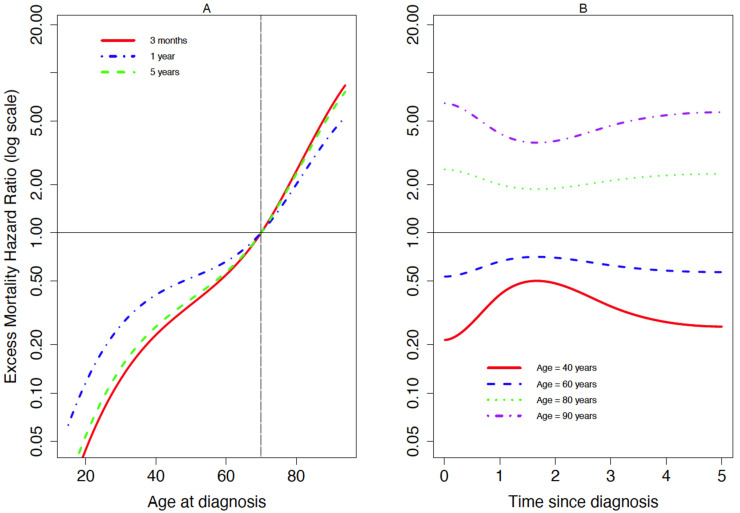
Excess mortality hazard ratios according to (**A**) age at diagnosis at different time since diagnosis (3 months, 1 and 5 years), and (**B**) time since diagnosis for different age groups, amongst patients diagnosed with follicular lymphoma (*n* = 15,516) in England during 2005–2013.

**Figure 3 cancers-13-05805-f003:**
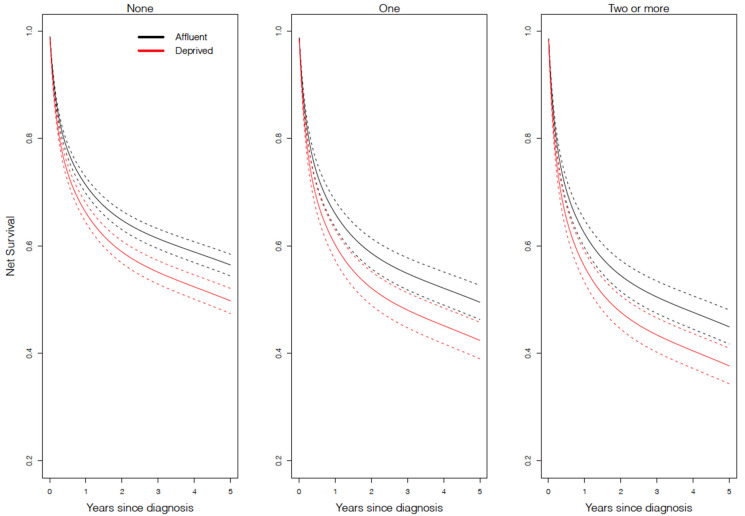
Net survival model-based prediction for diffuse large B-cell lymphoma for each comorbidity status by deprivation level (*n* = 29,898) in England between 2005 and 2013. The values here are presented for 70-year-old white males diagnosed via a general practitioner referral. Values will change for other covariates, but the pattern observed here will remain.

**Figure 4 cancers-13-05805-f004:**
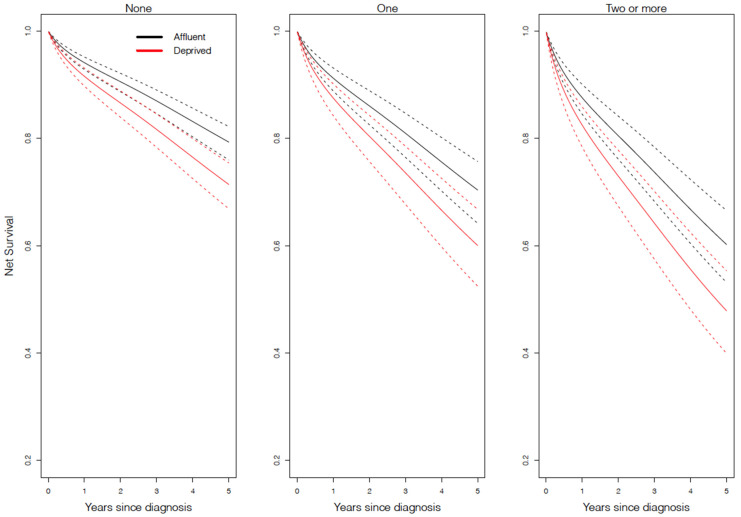
Net survival model-based prediction for follicular lymphoma for each comorbidity status by deprivation level (*n* = 15,516) in England between 2005 and 2013. The values here are presented for 70-year-old white males diagnosed via a general practitioner referral. Values will change for other covariates, but the pattern observed here will remain.

**Table 1 cancers-13-05805-t001:** Distribution of cancer subtypes by patient and healthcare system characteristics for patients (*n* = 45,414) diagnosed with non-Hodgkin lymphoma in England during the period 2005–2013.

Patient Characteristics	Subtype of NHL			
	FL		DLBCL	
	*N* = 15,516		*N* = 29,898	
Age (mean, SD)	63.9 (13.6)		67.4 (14.9)	
Sex, *n* (%)				
Male	7318	(47.2%)	16,215	(54.2%)
Female	8198	(52.8%)	13,683	(45.8%)
Deprivation quintiles (Q), *n* (%)				
Least deprived (Q1)	3547	(22.9%)	6340	(21.2%)
Q2	3517	(22.7%)	6663	(22.3%)
Q3	3294	(21.2%)	6246	(20.9%)
Q4	2925	(18.9%)	5863	(19.6%)
Most deprived (Q5)	2233	(14.4%)	4786	(16.0%)
Comorbidity status, *n* (%)				
No comorbidity	14,343	(92.4%)	26,718	(89.4%)
One comorbidity	641	(4.1%)	1570	(5.3%)
Multimorbidity	532	(3.4%)	1610	(5.4%)
Route of diagnosis, *n* (%)				
GP referral	6297	(44.0%)	8157	(28.7%)
A&E	1869	(13.1%)	9617	(33.8%)
Secondary care	2222	(15.5%)	3724	(13.1%)
TWW	3912	(27.4%)	6918	(24.4%)
Missing *	1216	(7.8%)	1482	(5.0%)
Ethnicity, *n* (%)				
White	11,052	(94.9%)	21,739	(94.1%)
Others	600	(5.2%)	1369	(5.9%)
Missing *	3864	(24.9%)	6790	(22.7%)

GP: general practitioner referral, A&E: accident and emergency room, TWW: two-week-wait. Complete case analysis: missing ethnicity 23.5%; missing route to diagnosis 5.9%. * Proportions are of the total number of patients.

**Table 2 cancers-13-05805-t002:** Adjusted excess mortality hazard ratios for age, sex, deprivation, comorbidity, cancer subtype, route of diagnosis, ethnicity, and LSOA as random intercept for (i) complete case analysis, and (ii) after multiple imputation for patients (*n* = 29,898) diagnosed with diffuse large B-cell lymphoma in England during the period 2005–2013.

Patient Characteristics	Model (i): Complete Case			Model (ii): After Imputation		
	HR	CI	*p*–Value	HR	CI	*p*–Value
Sex						
Male	Ref	Ref		Ref	Ref	
Female	0.93	0.89–0.98	0.003	0.93	0.90–0.96	<0.001
Ethnicity						
White	Ref	Ref		Ref	Ref	
Other	0.97	0.87–1.08	0.556	0.99	0.91–1.08	0.809
Deprivation quintiles (Q)						
Least deprived Q1	Ref	Ref		Ref	Ref	
Q2	1.03	0.96–1.11	0.372	1.00	0.93–1.08	0.922
Q3	1.08	1.00–1.16	0.045	1.07	1.00–1.14	0.045
Q4	1.17	1.08–1.26	<0.001	1.13	1.04–1.23	0.003
Most deprived Q5	1.26	1.16–1.37	<0.001	1.22	1.18–1.27	<0.001
Comorbidity status						
No comorbidity	Ref	Ref		Ref	Ref	
One comorbidity	1.26	1.15–1.38	<0.001	1.23	1.14–1.32	<0.001
Multimorbidity	1.50	1.38–1.64	<0.001	1.40	1.01–1.94	0.043
Route of diagnosis						
GP referral	Ref	Ref		Ref	Ref	
A&E	2.75	2.60–2.91	<0.001	2.75	2.54–2.98	<0.001
Secondary Care	1.43	1.22–1.67	<0.001	1.23	1.11–1.36	<0.001
TWW	1.33	1.23–1.45	<0.001	0.83	0.56–1.24	0.362
Random Effect						
SD (SE)	0.48 (0.08)	-	-	0.39 (0.04)	-	-

GP: general practitioner referral. A&E: accident and emergency room. TWW: two-week-wait.

**Table 3 cancers-13-05805-t003:** Adjusted excess mortality hazard ratios for age, sex, deprivation, comorbidity, cancer subtype, route of diagnosis, ethnicity, and LSOA as random intercept for (i) complete case analysis, and (ii) after multiple imputation for patients (*n* = 15,516) diagnosed with follicular lymphoma in England during the period 2005–2013.

Characteristics	Model (i): Complete Case			Model (ii): After Imputation		
	HR	CI	*p*–Value	HR	CI	*p*–Value
Sex						
Male	Ref	Ref		Ref	Ref	
Female	0.86	0.76–0.96	0.010	0.89	0.81–0.97	0.009
Ethnicity						
White	Ref	Ref		Ref	Ref	
Other	0.59	0.41–0.83	0.003	0.76	0.60–0.96	0.019
Deprivation quintiles (Q)						
Least deprived Q1	Ref	Ref		Ref	Ref	
Q2	1.09	0.91–1.31	0.364	1.10	0.92–1.32	0.309
Q3	1.23	1.02–1.48	0.030	1.11	0.96–1.29	0.166
Q4	1.37	1.13–1.65	0.001	1.34	1.06–1.69	0.015
Most deprived Q5	1.69	1.38–2.06	<0.001	1.45	1.30–1.62	<0.001
Comorbidity status						
No comorbidity	Ref	Ref		Ref	Ref	
One comorbidity	1.51	1.19–1.91	<0.001	1.52	1.25–1.84	<0.001
Multimorbidity	2.38	1.90–3.00	<0.001	2.19	1.45–3.31	<0.001
Route of diagnosis						
GP referral	Ref	Ref		Ref	Ref	
A&E	3.18	2.69–3.76	<0.001	3.32	2.49–4.43	<0.001
Secondary Care	1.27	0.86–1.90	0.233	1.22	0.96–1.55	0.107
TWW	1.17	0.98–1.40	0.084	1.06	0.63–1.78	0.830
Random Effect						
SD (SE)	0.87 (0.14)	-	-	0.69 (0.16)	-	-

GP: general practitioner referral. A&E: accident and emergency room. TWW: two-week-wait.

## Data Availability

The data that support the findings of this study are available via application to the Public Health England Office for Data Release, but restrictions apply to the availability of these data.

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
