# Peer review of "Excess Mortality by Multimorbidity, Socioeconomic, and Healthcare Factors, amongst Patients Diagnosed with Diffuse Large B-Cell or Follicular Lymphoma in England"

_cancers, 2021, doi:10.3390/cancers13225805_

Round 1
Reviewer 1 Report
Multiple factors (age, comorbidity, socioeconomic factors, etc.) must influence the treatment for malignant lymphoma (i.e., intensive or mild chemotherapy or watch and wait (especially in case of FL), with or without rituximab, number of chemotherapy course completed, etc). Therefore, authors need to incorpolate data regariding treatment into the hazard ratio analysis.
Author Response
Please kindly see the attached response letter.

Reviewer 2 Report
Matthew J. Smith et al. uncovered cancer epidemiology from the standpoint of approaching the pts with Diffuse Large B-cell Lymphoma and Follicular Lymphoma.
Points to be considered:
- One issue related to the strategy employed can be the lack of fully control for confounding by indication leading to model misspecification. This would be an issue even with a perfectly robust model. This confounding is best illustrated by the unexpected results for chemotherapy and immunological treatment: the variation explained by these models may be great, because they allow the effect of time‐varying variables to be modelled and, hence, measures of prognostic factors that are updated over time since the cancer diagnosis. Can the author comment on this?
- Since novel strategies are being developed in treating NHLs (refer to PMID: 26818572) how would the authors suggest to overcome the non clinical barriers to those approaches (i.e. BiTEs, CAR-T, ADCs, etc.)?
3. The underlying message here is that more precision and individualized approaches need to be tested in well designed clinical trials – a challenge, but I would be interested in their perspective of how this might be done.
Author Response
Please kindly see the response letter attached.

Round 2
Reviewer 1 Report
The revised manuscript has been improved.
Reviewer 2 Report
The authors have clarified several of the questions I raised in my previous review. Most of the major problems have been addressed by this revision.